# Indoor Air Radon Concentration in Premises of Public Companies and Workplaces in Latvia

**DOI:** 10.3390/ijerph19041993

**Published:** 2022-02-10

**Authors:** Jelena Reste, Ilona Pavlovska, Zanna Martinsone, Andris Romans, Inese Martinsone, Ivars Vanadzins

**Affiliations:** 1Institute of Occupational Safety and Environmental Health, Riga Stradins University, Dzirciema Street 16, LV-1007 Riga, Latvia; jelena.reste@rsu.lv (J.R.); zanna.martinsone@rsu.lv (Z.M.); ivars.vanadzins@rsu.lv (I.V.); 2Laboratory of Hygiene and Occupational Diseases, Riga Stradins University, Ratsupites Street 5, LV-1069 Riga, Latvia; inese.martinsone@rsu.lv; 3Radiation Safety Centre State Environmental Service of the Republic of Latvia, Rupniecibas Street 23, LV-1045 Riga, Latvia; andris.romans@vvd.gov.lv

**Keywords:** indoor air quality, radon gas, radon mapping, prone areas, air exchange, public areas, workplaces, building materials

## Abstract

Considering the multitudes of people who spend their time working indoors in public premises and workplaces, it is worth knowing what their level of exposure is to natural radioactive radon gas, the second most widespread and dangerous carcinogen for lung cancer development after cigarette smoking. This state-level study covered most of the territory of Latvia and conducted 941 radon measurements with Radtrack2, placed for 4–6 months in the premises of public companies, educational institutions, medical care institutions, etc. The study found that 94.7% of samples did not exceed the national permissible limit (200 Bq/m^3^), the level at which preventive measures should be initiated. The median value of average specific radioactivity of radon in these premises was 48 Bq/m^3^ (Q1 and Q3 being 27 and 85 Bq/m^3^), which is below the average of the European region. Slightly higher concentrations were observed in well-insulated premises with plastic windows and poorer air exchange, mostly in schools (59 (36, 109) Bq/m^3^) and kindergartens (48 (32, 79) Bq/m^3^). Industrial workplaces had surprisingly low radon levels (28 (16, 55) Bq/m^3^) due to strict requirements for air quality and proper ventilation. Public premises and workplaces in Latvia mostly have low radon concentrations in the air, but more attention should be paid to adequate ventilation and air exchange.

## 1. Introduction

According to the list of cancerogenic compounds by the World Health Organization (WHO) and International Agency for Research on Cancer (IARC) reports, radon gas has been acknowledged as the second major cause of lung cancer after tobacco smoking [1]. The permanent presence of radon in the air indoors can put people at risk for the development of airway malignancies. Considering the time spent indoors today by people both at work and during leisure periods, indoor air quality has an enormous impact on human health. Workers in some occupations can be exposed to radon at higher levels when performing tasks underground or in tight spaces located close to ground level [2]. 

The history of radon began in the 15th century when a high mortality from lung diseases was observed among young miners in Germany [3]. Although many more radon measurements were performed in these mines to support the hypothesis, it was not generally accepted until 1953, when, finally, it was realised that radon was the cause of these lung cancers. The first radon measurements in certain premises were carried out by a Swedish researcher, *Bengt Hultqvist*, in 1956 and were launched and carried out at the same time in several countries [4].

In 1988, an indoor radon reduction law was signed by the Environmental Protection Agency (EPA), which set a long-term target for indoor air to be radon free. The law authorised the funding of radon-related activities at the state and federal level, providing national programmes and supplying technical assistance, radon surveys in schools, federal buildings and training centres and a qualification programme for companies providing radon services, as well as developing a citizen’s manual on radon and model design standards. In 2009, the EPA set guidelines for maximum environmental radon levels to limit the risk of developing lung cancer from radon exposure. Lately, radon issues have interested scientists, politicians, labour leaders, and the public [5].

The radon isotope (^222^Rn or radon-222) is an inert, radioactive gas with no colour or odour and is 7.5 times heavier than air. The half-life period of ^222^Rn is 3.8 days. Radon gas is formed by the decay of radium (^226^Ra or radium-226). Radon is highly soluble in water and organic liquids. It is a product of the degradation chain of existing natural uranium (^238^U or uranium-238) [6,7,8]. Radioactive decay of radon forms alpha radiation that has high radiotoxicity. The main sources of radon gas are in subterranean depths. Specifically, the geological structure of the site determines the risk of increased radon in the human environment. The natural origin of radon comes from rock salts or sediment, such as clay [9]. High radon concentrations can occur near the earth’s surface but not above the second or third floor of a residential building. Radon is also found in various building materials such as granite, marble, concrete, cement, and bricks (especially clay) [8]. Higher radon concentration is possible if these materials are used in building construction.

Radon gas can escape any chemical compound and easily diffuses through the ground into the air. It can enter buildings through cracks, pores and cavities in the basement or around the water supply system, resulting in gas collection indoors. Most of the population’s exposure to radon occurs at home [7]. Considering the time spent indoors by most people nowadays, especially during the COVID-19 disease pandemic, indoor air quality issues have become an urgent question for public health. 

Every nucleus of ^222^Rn eventually decays through the sequence of isotopes: polonium-218 (^218^Po), lead-214 (^214^Pb), bismuth-214 (^214^Bi), polonium-214 (^214^Po), and lead-210 (^210^Pb). With each transformation along this path, the nucleus emits characteristic radiation: alpha particles, beta particles, gamma rays, or combinations of these [10]. When the radon nucleus decays, it releases an alpha particle with 5.49 MeV of energy, and the nucleus transforms to ^218^Po. Polonium atoms are metals and tend to stick to surfaces they come in contact with, e.g., a dust particle in the air, a wall, or the inside of the lungs [11,12,13].

The alpha particles emitted from the short-lived decay products of radon can severely damage the airway epithelium. That is why it is the second leading cause of lung cancer in humans after cigarette smoking. However, there is no known threshold concentration below which no risk exists [11,12,14,15]. Radon alpha emissions in the long term can have an adverse effect on the health of the population. In humans, radon gas enters the body through the air by inhalation or by drinking water with dissolved radon gas in it. Low radon levels do not harm the human body, but the effects of prolonged radon exposure contribute to undesirable changes, primarily in the respiratory and digestive organs. Radon’s adverse health effects are significantly higher in smokers, with an added risk of respiratory disease. Radon alpha radiation may increase the risk of developing lung cancer and reduce a person’s life span. Therefore, it is important to find the average radon specific radioactivity in a building so that, if necessary, proper measures for reducing the radon level can be undertaken [14].

Measurements of indoor radon concentrations in dwellings and workplaces have been conducted in many countries [13]. A recent focus has been on the variations in indoor radon concentrations, which depend on many factors such as the geological area and soil properties, as well as seasonal variations. Some scientific reports focused on the characteristics of the building, such as room types, floor levels, ventilation systems, building materials, type of building basement, type of windows (wooden, plastic, etc.), type of plastering (insulation), building age, and others [16,17,18,19,20,21,22,23,24,25,26]. Because of the many factors that influence indoor radon concentrations, only real measurements of the radon concentrations can predict the radon hazard and supply a correct risk assessment. 

In light of the recommendations of world organisations, studies at a national level have been launched according to Directive 2013/59/Euratom to assess whether planning additional protective measures against radon are necessary for Latvia. 

The primary objectives of the Basic Safety Standards (BSS) Directive 2013/59/Euratom and the Environmental Policy Guidelines for 2014–2020 identified the need to carry out an assessment on the state level and perform measurements in the whole territory of Latvia. The project was supported by the International Atomic Energy Agency (IAEA), and radon measurements were indeed conducted, including information about building age and type, renovation and insulation, type of windows, ventilation systems, and other parameters.

### Review of Previously Estimated Risk of Radon Gas in Latvia

Radon gas liberated into the environment can be a source of radioactive pollution. The main factors for the appearance of radon gas in the environment are the geological and granulometric composition of Quaternary deposits [17]. Considering the geological structure, radon propagation problems in Latvia could be regarded as non-essential. However, as we found in our measurements, there still might be places in Latvia where there is an increased amount of radon in buildings.

The study on the initial assessment of radon in the territory of Latvia was organised by the Ministry of Environmental Protection and Regional Development (MEPRD) of Latvia and fulfilled by the geological research company “Geo Consultants, Ltd.” in 2014–2015. In this study, the radon risks in the country’s territory were predicted based on geological mapping data (Figure 1). According to these theoretical geological maps, territories with a locally increased risk of radon, a normal probable risk of radon and a negligible risk of radon have been isolated in Latvia [27,28].

## 2. Materials and Methods

### 2.1. Institutions Involved

According to authorities, it is relevant to know the radon gas concentration levels in indoor air. Extensive research had not been conducted in Latvia before; therefore, in 2016, the Radiation Safety Centre State Environmental Service of the Republic of Latvia (RSC SES) with the support of the Riga Stradins University (RSU) agency Institute of Occupational Safety and Environmental Health (IOSEH) launched the project “Radon gas measurements in Latvia’s workplaces and public buildings” in 2016–2017. The project was supported by the International Atomic Energy Agency (IAEA) Technical Cooperation Program (TCP), providing measurement services and detectors. Initially, identification of possible radon measurement sites was carried out from September to October 2016 by IOSEH, in cooperation with the RSC SES and the MEPRD.

### 2.2. Course of Study

The RSC SES, in cooperation with the contact person appointed by the company/institution between November and January, placed 955 passive radon gas detectors in various workplaces and public buildings. Overall, detection of radon gas was performed in 100 out of 119 municipalities of Latvia, i.e., approximately 83% of the terrestrial area (Figure 2).

The 955 passive radon gas detectors were deployed at 197 institutions/companies (pre-schools, schools, and different workplaces such as hospitals, museums, cafeterias, manufacturing, and service companies). The detectors were placed inside 243 buildings exclusively on the first and ground floor, where most employees/residents were located. On average, detectors were placed in four locations in each building of the company/institution. The measurements of radon gas lasted for 4–6 months in each selected building. Additionally, the specially designed protocol was followed for every measurement site to ensure careful data analysis after retrieving the detectors. The wide range of indoor radon gas concentrations was investigated according to several types of building materials, building age, type of windows and ventilation habits and system used.

Fourteen detectors out of 955 (1.5%) were considered lost or damaged, resulting in 941 detectors valid for analysis. After removing the detectors, the RSC SES organised their dispatch to a laboratory in Sweden accredited in accordance with ISO 11665-4 which read the measurements of the radon gas level. Obtained data were analysed in context with the protocols.

### 2.3. Measurement Equipment

Radtrack2 alpha track detectors (Radonova Laboratories AB, Long Term Radon Test) were used to measure indoor radon concentrations in the current study. Radtrack2 uses a continuous passive radon sampling and delayed analysis method. The operation of these detectors is based on passive entrapment of radon gas inside the anti-static housing of the device and on fixing tracks of alpha particles on the film during the measurement period. The device consists of film elements in an anti-static plastic housing. Radon enters the detector by diffusion, and tracks of emitted alpha particles are captured by film. Readings from the film represent the summary outcome of radon in the premises during the sampling time. The acquired data are then processed and recalculated according to the duration of measurements and the size of the premises. All results are given in a specific numerical value of Becquerels per cubic metre (Bq/m^3^). The alpha track is the most commonly used method for measuring radon in the world. Radtrack2 has been reported as a reliable detector for measurement purposes.

### 2.4. Criteria for the Placement of Detectors

The measurement locations were selected with the reasonable expectation that the measurement device would not be disturbed during the long measurement period. The areas where occupants spent most of their time were selected for the measurements. Radon measurement was conducted in each occupied basement room or, if no basement existed, on the ground floor or the lowest-level floor with occupied rooms. A room was defined as a space enclosed by walls that reach the ceiling, and a room subdivided by partitions was treated as one room. An occupied room was defined as one in which an individual spends more than four hours per day.

The preferred location for the radon detectors was at a height from 0.8 m to 2 m from the floor (typical breathing zone), approximately 40 cm from an interior wall or approximately 50 cm from an exterior wall, at least 50 cm from the ceiling, and 20 cm from other objects to allow normal airflow around the device. To ensure accurate measurements, the devices were not placed in air currents caused by heating, ventilating, and air conditioning vents, doors, fans and windows. The detectors were not placed on or near electrically powered appliances or other equipment such as computers, television sets, or speakers, or near heat sources, such as over radiators, near fireplaces or in direct sunlight, because the measurement devices could be damaged, or the measurements affected. Measurements were not done in bathrooms, closets, cupboards, etc., or at upper levels of the building as radon concentrations in these areas might not be representative. In a complex of buildings, such as a hospital, every building was tested separately.

For a representative estimate of school radon levels, measurements were conducted in occupied classrooms or offices located on the lowest level of the building. For larger rooms, one detector was placed for every 200 m^2^ of floor space.

### 2.5. Data Processing and Statistics

After the measurement readings were received from the certified laboratory, they were combined with protocol data about placement sites and later were processed together. Sensitive information about measurement places was recoded to avoid direct identification. Data were grouped by the type of premises, building material, ventilation system, aeration mode, etc., and compared to each other. Data were tested for normality of distribution. For data with non-normal distribution, nonparametric methods (e.g., Chi-square test, Mann–Whitney test, Spearman’s correlation) were used for analysis. Computer programs Microsoft Excel Professional Plus 2010 and IBM SPSS Statistics version 20 were used for calculations. Differences were considered statistically significant at *p* < 0.05.

## 3. Results

As mentioned previously, the current study was designed to obtain knowledge about radon levels in public buildings in Latvia, evenly covering the whole territory of the country. For this purpose, 735 passive radon detectors were placed on the 1st floor and basement rooms of educational institutions (303 in pre-schools, 397 in secondary schools and 35 in high schools) and 202 in other workplaces, in the rooms where employees were most often present. Other workplaces where detectors were placed included: hospitals (48 detectors), food enterprises (29), museums (9), and other public places (116) such as small commercial premises, labs, cafeterias, etc. (Table 1). In total, the variations of radon concentration were from 1 to 460 Bq/m^3^, with the median value 48 Bq/m^3^ (Q1 and Q3 being 27 and 85 Bq/m^3^, respectively). Overall, 94.7% of all samples were below 200 Bq/m^3^, i.e., below the national permissible level for indoor air in buildings. There were only 3 extremes that were excluded from these calculations (681, 774 and 1341 Bq/m^3^). For every extreme value, a detailed analysis was performed (described later in this paper).

A significant difference in radon concentration levels between educational institutions and workplaces was found (*p* < 0.05). The median concentration of radon in the air of educational institutions was almost double that at other enterprises, 54 (31, 91) and 28 (16, 55) Bq/m^3^, respectively (*p* < 0.001). The highest median value was found in the air of primary and secondary schools, 59 (36, 109) Bq/m^3^ (Table 1). The next highest median value was observed in kindergartens, 48 (32, 79) Bq/m^3^. Another high median level of radon which might affect a large number of very sensitive people was found in hospitals, 36 (19, 64) Bq/m^3^. Surprisingly, in the premises related to higher education, the radon concentrations were at the lowest level among all the explored enterprises; the median value was 24 (12, 44) Bq/m^3^. On the contrary, museums showed high concentrations of radon (median value 49 (32, 138) Bq/m^3^), but the number of measurements was limited (only 9). In addition, the number of people potentially exposed to radon in those locations and the length of exposure might be lower. A graphical summary of these data is shown in Figure 3.

When analysing radon concentration in public buildings by territorial location in Latvian regions, it can be concluded that significant differences among regions are present (*p* < 0.001, Figure 2). The highest median radon concentration was found in Rauna municipality (352 (208, 439) Bq/m^3^) and the lowest was in Broceni municipality (5 (3, 13) Bq/m^3^). The median values of radon concentration by Latvian municipalities are summarised in the Appendix A.

The effect of building material on indoor radon concentration was not clearly seen in the current study, as combinations of different materials in the construction of a single building are quite popular and can be hardly recognised, especially in old and reconstructed buildings. The median concentrations of radon by dominant building material were as follows: bricks: 56 (29, 98) Bq/m^3^ (*n* = 416);stone: 50 (28, 92) Bq/m^3^ (*n* = 182);concrete: 43 (25, 72) Bq/m^3^ (*n* = 267);other materials (metal, expanded clay aggregate, etc.): 42 (25, 64) Bq/m^3^ (*n* = 48).

The presence or absence of a cellar minimally affected the indoor radon levels. In buildings with a cellar, the median radon level was 47 (27, 86) Bq/m^3^ (*n* = 698), while without a cellar, it was slightly higher (52 (27, 85) Bq/m^3^, *n* = 239, *p* > 0.05). 

On the other hand, the most influencing factor was found to be the type of windows used in the premises. In the presence of plastic windows (Figure 4), radon levels were even higher than in premises without windows, 52 (30, 88) Bq/m^3^, *n* = 750 vs. 42 (17, 80) Bq/m^3^, *n* = 96, *p* = 0.005. Logically, wooden windows with better air penetration showed lower levels of radon, 38 (15, 62) Bq/m^3^, *n* = 49. The difference in radon levels in locations with plastic and wooden windows was statistically significant (*p* = 0.003). 

The presence of insulation influenced the level of radon as well. In insulated buildings, the level of radon was higher (52 (29, 88) Bq/m^3^, *n* = 677) than in non-insulated buildings (41 (22, 76) Bq/m^3^, *n* = 252, *p* < 0.001).

The presence of mechanical ventilation showed lower concentrations of radon indoors (45 (23, 81) Bq/m^3^, *n* = 317), while in locations with natural ventilation, higher levels of radon were observed (54 (32, 93) Bq/m^3^, *n* = 432). In the case of combined ventilation (natural and mechanical), radon levels (52 (29, 84) Bq/m^3^, *n* = 82) were similar to levels in locations with natural ventilation, probably indicating rare and improper use of mechanical ventilation. It is important to note that natural ventilation was mostly found in educational institutions such as kindergartens and secondary schools, while mechanical ventilation was mostly found in workplaces at enterprises, high school premises and hospitals. This might explain the major differences in radon levels between educational institutions and enterprises. Considering aeration regimens, the highest median concentration of radon was found in premises aerated for only 0.5 to 3 h daily (52 (31, 88) Bq/m^3^, *n* = 649 or 73.2% of all analysed samples) or 3–8 h daily (46 (26, 94) Bq/m^3^, *n* = 113 or 12.7%).

On the other hand, logically, the lowest median radon concentration was observed in premises aerated for more than 8 h per day (38 (18, 57) Bq/m^3^, *n* = 52), but this occurred in only 5.9% of all the analysed samples. In total, short-term aeration (up to 3 h) was most common (39% of all analysed premises), especially among those premises with natural ventilation (81%). As mentioned previously, natural ventilation was found mainly in schools (51% of all schools included) and kindergartens (53.5% of all kindergartens evaluated), which was 83% of all the premises included in the study. Additionally, it is worth mentioning that the highest percentage of insulated buildings and installed plastic windows were among schools (80.9% insulated and 89.5% with plastic windows) and kindergartens (80.3% insulated and 94.3% with plastic windows). All this proves that the main reason for increased concentration levels of radon in these institutions was inadequate air exchange due to reduced natural ventilation and decreased aeration time. 

Only 5.3% (*n* = 50) of all samples indicated an indoor air radon concentration above 200 Bq/m^3^. At times the increased levels were found because of placing several detectors in different locations in one building, e.g., in 33 buildings out of the 50 highest levels measured, at least one radon gas measurement was higher than 200 Bq/m^3^. The results of the study highlight that elevated radon concentrations were observed mainly in:wardrobe rooms without windows and mechanical ventilation in educational institutions;auxiliary rooms, where workers stay temporarily and premises are not ventilated regularly;library premises that are rarely aerated;buildings with insulation that reduced natural air exchange;buildings that were insulated with coal slag.

In all the cases when the elevated radon concentration above 200 Bq/m^3^ was found, the RSC SES informed the organisations involved in the study about the results obtained. Recommendations for the reduction of radon levels were provided for them as well. Additionally, on-site indicative tests and detailed analysis of the situation were held in buildings with the highest radon scores, and guidance on possible measures to reduce radon levels was provided.

There were some extremes in the measurements that were further excluded from the calculations. The extremes are the highest or lowest values from the data set and therefore do not say anything about their nature. For example, the highest value, 1341 Bq/m^3^, was found in a kindergarten, where the detector was placed under the ceiling to exclude access to the detector by people or inadvertently touching the window during these 6 months. As we determined later, high levels of radon could be found because of the specific features of the building, this being historical building made from stone and renovated on the inside with gypsum plates and insulated on the outside with polystyrene facades, thus closing natural ventilation inside and resulting in a carbon dioxide increase in the room of more than 2300 ppm. This effect was also reinforced by the installation of new plastic windows. Finally, one detector was placed under the ceiling near the end of a curtain rod. The hole that was drilled for the curtain rod let in air between two new walls, isolating the old wall next to the detector. The result of the mistake in placing the detector was a high level of radon measured. Therefore, it is important to complete and check the instructions to avoid such unwanted deviations.

Another high value of 774 Bq/m^3^ in radon measurements was found in a specific workplace located close to the ground level without a cellar where the well and filters for centralised obtaining drinking water were found. The premises had poor ventilation, but the staff worked there for a limited time only. The presence of the natural water source coming from underground in combination with poor ventilation could be the reason for the increase in indoor radon concentration.

The third extreme excluded from the data analysis was 681 Bq/m^3^. It was found in a region naturally prone to higher concentrations of radon coming from the soil due to a specific geological structure (according to the previous geological assessment, Figure 1). This high measurement was obtained from a brick building with a cellar below and a wooden floor. The building was recently insulated with a polystyrene facade, and plastic windows were installed, but only natural ventilation was supported, commonly aerating for 3 h per day. A primary school was located there, but luckily the premises were used for administrative purposes and not for children. In the same building, radon was measured in other locations as well, but the levels there were only slightly increased (201 Bq/m^3^) or even normal (167 Bq/m^3^ and 73 Bq/m^3^), but still more than in other Latvian regions. In this case, recent insulation that reduced natural air exchange, poor ventilation, and geographic location played the main roles in the increase in radon level.

Considering the non-normal distribution of the data on radon concentration in the air acquired in the study with high positive skewness (skewness S_sk_ = 2.37, standard error SE = 0.08; kurtosis S_ku_ = 7.17, SE = 0.16), a lognormal quantile–quantile (Q–Q) plot was analysed. A lognormal Q–Q plot is a graphical method that helps to look at the distribution of any random variable. It is a statistical approach to observe the nature of any distribution and compare the unknown distribution with the known normal one.

The central part of the Q–Q plot data falls along a straight line, but there are deviations from the expected distribution both at the upper (heavy-tailed) point and lower (light-tailed) point. It can be assumed that the values are, for the most part, normally distributed (see Figure 5). Despite the straight midline, the lagging tails may indicate that the acquired results are higher than expected (both minimum and maximum in data) [29,30].

The deviation from log-normality is often stronger in the light-tail and is of no interest because it does not correspond to a radiological problem. Therefore, it is worth considering deviations in the heavy-tailed part of the distribution above the median [31].

## 4. Discussion

Humans are constantly exposed to low doses of ionising radiation from several natural sources. Natural radiation mainly comes from cosmic origins, from the ground, and from buildings, while radioactivity within the human body enters through air, food and water. Radon (including thoron) amounts to about 52% of the total ionising radiation we are exposed to; cosmic rays make up 16%, food 12%, terrestrial inputs 20% and about 3% comes from other sources. At the same time, ionising radiation includes all human-made (artificial) sources [32].

Manufactured sources can arise from peaceful purposes (e.g., medical use, energy generation, associated fuel cycle facilities and waste management) and military purposes [33].

Radon gas is a natural radiation source. Radon gas enters from the soil beneath buildings through cracks and openings in basements. The air pressure inside these premises is, from time to time, lower than the pressure in the soil under the building. As a result of this difference in pressure, the building acts like a vacuum, drawing the radon from the soil. Typically, cracks and openings are found in places where the floor meets the wall, expansion connections to the floor, openings in the floor for pipes and wires, and hollow walls that enter the floor. 

All the natural sources mentioned previously form natural background radiation with certain levels characteristic for a particular region (exposing inhabitants to around 2.4 mSv per year; 1.2 mSv per year comes from radon and its decay products). Natural background radiation levels can vary greatly depending on soil composition and other terrestrial characteristics [33]. The exposure to ionising radiation exceeding common background levels should not exceed a dose of 1 mSv/year as the annual dose limit for the general public. Considering various occupational exposures to radiation, the dose limit for professionals is 20 mSv/year, and a 6 mSv limit has been set as the largest dose for students or trainee workers [10].

There must be an existing exposure situation in the case of radon entering indoor workplaces since the presence of radon does not depend to a significant extent on human activities carried out at the workplace [34].

Radon levels change all the time and depend on the season. To obtain a more precise sense of radon exposure, the best solution is to place detectors for a period measuring at least 4–6 months; in this case, the results represent the average annual value of radon gas exposure. The detector must be placed at a breathing level height to make sure that the measured radon level is the radon in inhaled air. Radon may accumulate to high concentrations in closed or poorly ventilated indoor spaces and enclosed underground spaces. Radon levels are usually the highest in basements and crawl sites because radon is heavier than air. In addition, these zones are often closer to the source of radon and are usually poorly ventilated. Measurements in buildings that do not have a central air conditioner during warm weather may result in misleading results, as it is highly likely that the windows will be open during measurements. This problem can be mitigated by increasing the duration of the test and emphasising the importance of long-term radon measurements. Re-testing should always be considered whenever a major renovation is carried out, which can significantly change the ventilation or airflow in the building or the use of spaces at the lowest borrowing level. In the case of major changes, a 3-month test should be carried out during the first heating season following the completion of the renewal [35].

“Regulations for Protection against Ionising Radiation”, Legal Act Nº149 (9 April 2002) of the Republic of Latvia [36], requires that, in situations where radon average specific radioactivity in a building, apartment or workplace is greater than 200 Bq/m^3^ per year, the owner of the premises together with specialists of the Radiation Safety Centre decide which protection measures need to be taken. Measures are mandatory if the average specific radioactivity of radon in the premises exceeds 1000 Bq/m^3^ at the time of the measurement or 600 Bq/m^3^ on average per year. In situations where the average activity of radon in underground or aboveground workplaces is higher than 400 Bq/m^3^ per year, employers should undertake protective measures against the harmful effects of radon, and the employment of pregnant women in those workplaces should not be allowed throughout the entire pregnancy.

Responsible institutions should ensure that employees at these workplaces are informed about the risks. In workplaces where workers’ exposure may exceed the effective dose of 6 mSv per year or a corresponding time-integrated radon exposure value, employees must be managed like in case of a planned exposure situation, and appropriate dose limits should be applied. Employers should determine which operational protection requirements need to be applied in certain situations.

The workplace must be designed and decorated to supply adequate protection or prevention against radiation. All radiation exposure should be kept as low as practical. The only way to know if elevated radon levels are present is to test. For instance, considering the large number of people who spend time in educational institutions which are simultaneously workplaces and public premises, the EPA recommends testing all schools for radon to ensure proper protection [37]. School boards should always consider re-testing whenever significant refurbishments are carried out, which can significantly change the ventilation or airflow in the building or the use of spaces below ground level.

The current study was designed to continue evaluating radon levels in Latvia. The SES under MEPRD conducted radon measurements in public buildings and workplaces with the project support of the IAEA Technical Cooperation Programs. Evaluating the radon measurements was carried out with passive detectors in public buildings and workplaces and areas presenting elevated radon gas levels were not found in many buildings. It can be concluded that Latvia has no areas with significantly elevated radon gas concentrations. In most cases (94.7% of samples), the average specific radioactivity of radon gas was below the specified level in buildings (200 Bq/m^3^). It has been concluded that elevated radon concentrations in buildings were caused by a variety of factors: the geological basis of the building, the building materials used for the building, or ventilation issues. The highest levels were observed in educational institutions (schools and kindergartens), where many young and sensitive people spend their time during the day, and the lowest numbers were found in workplaces such as food enterprises, kitchens or workshops. The explanation for this discrepancy might be that many schools and kindergartens in Latvia underwent renovations in the last years to improve energy efficiency, including installing plastic windows and insulation. Unfortunately, less attention was paid to improving air conditioning and ventilation systems. On the contrary, strict regulations about air quality exist for industrial and specific hazardous workplaces, which explains the better air exchange and lower numbers of radon levels. 

The radon concentrations in buildings above the limits can be reduced in several ways: by improving the ventilation of the building, by removing cracks in the structure of the building, as well as sealing walls and floors. Another way is improving underfloor ventilation or installing a system that allows the building to support positive air pressure. It is possible to use a separate set of measures on a case-by-case basis, so it is necessary to identify the radon’s path in the building and develop particular measures to reduce radon concentration. The simplest way to reduce radon concentration is to dwell in the premises regularly [38]. When buildings need to be insulated, it is necessary to evaluate and select the best solution to ensure the best possible indoor air exchange. Enclosed windows without passive ventilation create a risk of increasing radon concentration.

## 5. Conclusions

The current study evaluated indoor air radon concentration of public buildings and workplaces at the national level according to Directive 2013/59/Euratom in light of the recommendations of world organisations to assess whether planning additional protective measures against radon is necessary for Latvia. The indoor air radon gas concentration in most Latvian companies and public buildings is low and does not exceed the national permissible limit values. The data in our study indicate that higher levels of radon gas concentration were mainly attributable to insufficient ventilation (especially after the renovation of a building with insulation without improving the ventilation system). In order to reduce the concentration of radon gas in indoor air in Latvia, it is recommended to improve the quality of air exchange and ventilation, paying special attention to schools and pre-school educational institutions. 

## Figures and Tables

**Figure 1 ijerph-19-01993-f001:**
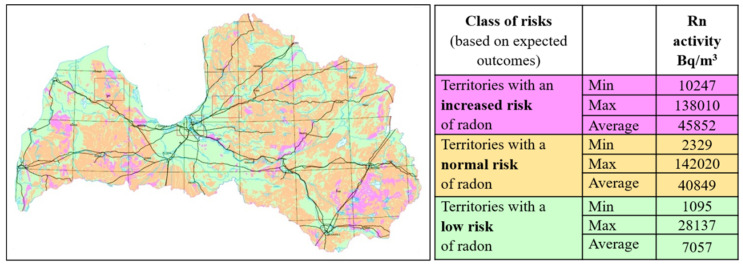
Geological radon risk forecast map in Latvia. The study was commissioned by MEPRD and performed by geological research company “Geo Consultants, Ltd.”.

**Figure 2 ijerph-19-01993-f002:**
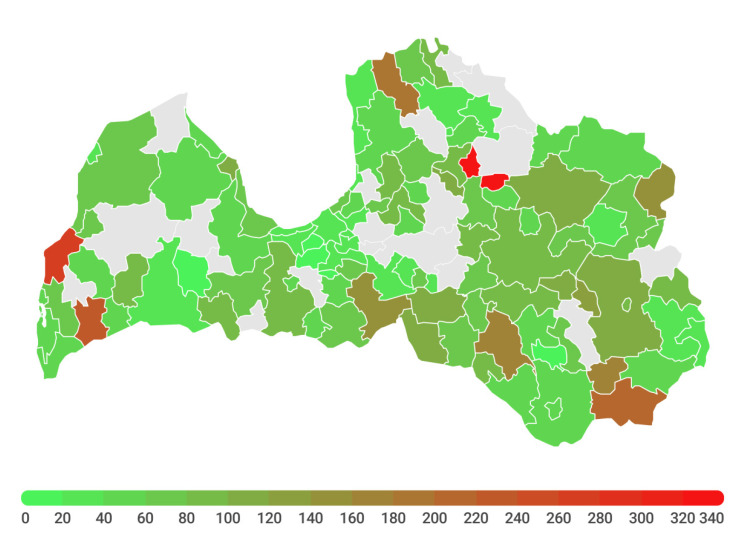
Median radon gas concentration in the indoor air in 100 administrative territories of Latvia (Bq/m^3^).

**Figure 3 ijerph-19-01993-f003:**
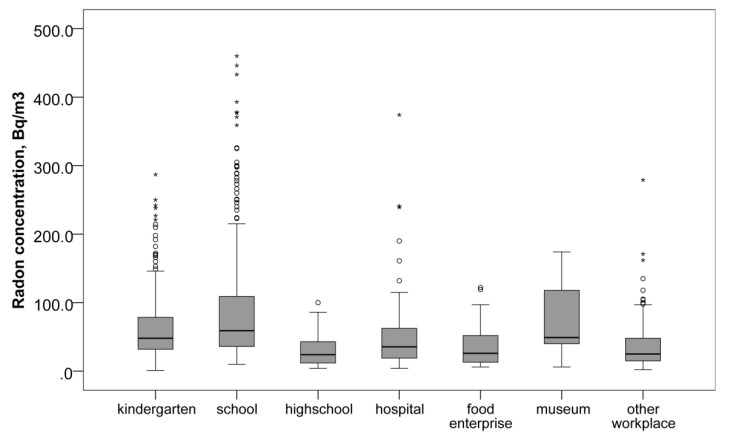
Radon concentration (Bq/m^3^) by type of public premises. * indicate extreme values.

**Figure 4 ijerph-19-01993-f004:**
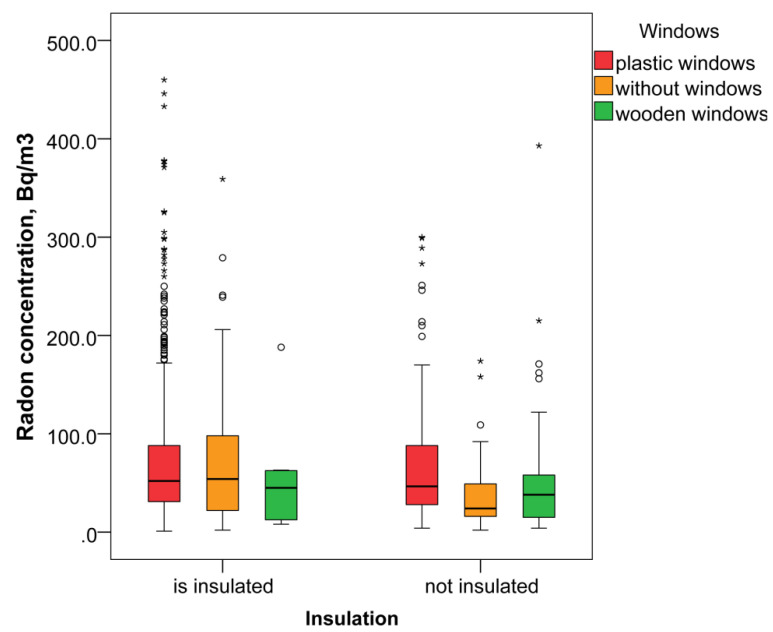
Radon concentration (Bq/m^3^) by type of windows and insulation done in public premises. * indicate extreme values.

**Figure 5 ijerph-19-01993-f005:**
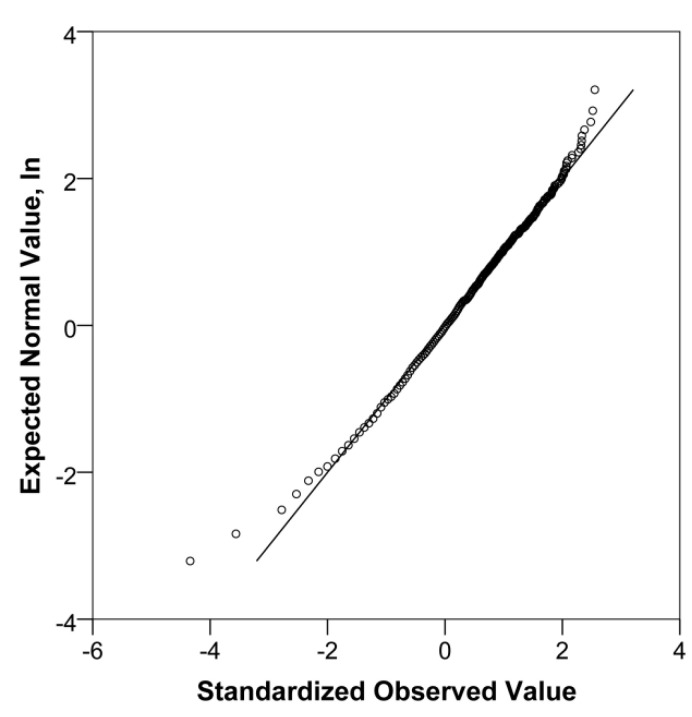
Q–Q plot of the radon concentration in premises of public companies and workplaces.

**Table 1 ijerph-19-01993-t001:** Radon concentration according to the type of workplace.

Type of Workplace	Number of Measurements	Radon Concentration, Bq/m^3^			
		Mean (± SD)	Median (Q1, Q3)	Min	Max
Kindergarten	303	62.75 (±46.75)	48 (32, 79)	1	287
School	397	88.48 (±80.26)	59 (36, 109)	10	460
Highschool	35	31.03 (±26.06)	24 (12, 44)	4	100
Hospital	48	60.13 (±72.62)	36 (19, 64)	4	374
Food enterprise	29	38.97 (±34.98)	26 (13, 53)	6	122
Museum	9	73.56 (±60.65)	49 (32, 138)	6	174
Other workplaces	116	40.54 (±42.08)	25 (15, 48)	2	279

## Data Availability

The data supporting reported results can be found at https://data.gov.lv/dati/lv/dataset/radona-gazes-limena-novertejums-latvija/resource/480e75c1-e3f8-49a8-bbe8-30842ec93e78.

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
