# Peer review of "Indoor Air Radon Concentration in Premises of Public Companies and Workplaces in Latvia"

_ijerph, 2022, doi:10.3390/ijerph19041993_

Round 1

Reviewer 1 Report

Dear authors,

please, read carefully muy suggestions contained in the attachment.

Keep the good work!

Author Response

Our manuscript has been carefully corrected according to your comments, objections, and suggestions. Please see the attached file.

Thank you!

Reviewer 2 Report

The paper is very interesting because it presents the results of monitoring the concentration of radon in Latvia with reference to national limit values.

The authors start from the objective reality according to which many people spend indoors  time in public premises and workplaces, so that  it is worth to know the level of exposure to natural radioactive radon gas, because it is the second most widespread and dangerous carcinogen for lung cancer development after cigarette smoking.

Detection of the radon gas was performed in 100 administrative territories out of total municipalities of Latvia, i.e., approximately 83% of terrestrial area.

Passive radon gas detectors were placed at 197 institutions/companies (pre-schools, schools and different workplaces such as hospitals, museums, cafeterias, manufacturing, and service companies) exclusively on the 1st and ground floor, where most people were located.

Because the radon level changes all over the time and depends on the season it is recommended to place detectors at least 4-6 months at a breathing level height to make sure that the measured radon level is the radon in an inhaled air. Even radon is accumulated in high concentrations in closed or poorly ventilated indoor spaces, as well  as in enclosed underground spaces, in basements and  crawl sites in zones placed closer to  the source of radon and  usually poorly ventilated,  radon levels in Latvia were below national norms.

The results obtained are very important for the management of the institutions where there are residents, visitors or employees but also for their education to ensure an adequate state of health. In this respect, responsible institutions should ensure that employees at these workplaces are informed about the risks. In workplaces, where workers' exposure may exceed the effective dose of 6 mSv per year or a corresponding time-integrated radon exposure value, employees must be managed as in case of a planned exposure situation and appropriate dose limits should be applied. Employers should figure out which operational protection requirements need be applied in certain situation.

We suggest to the authors if they have access to information regarding the health status of the people from the types of locations to make a correlation with the radon level and to monitor it in time. For kindergarten, school and highschool, subjects can be chosen who will spend at least 3 years in these institutions

Author Response

(The authors gave the same response as above.)
